# Multi-Criteria Evaluation of Site Selection for Smart Community Demonstration Projects

**Ming-Shiu Sung \*, Shen-Guan Shih and Yeng-Horng Perng**

Department of Architecture, College of Design, National Taiwan University of Science and Technology, Taipei City 106335, Taiwan; sgshih@mail.ntust.edu.tw (S.-G.S.); perng@mail.ntust.edu.tw (Y.-H.P.)
\* Correspondence: misuui@hotmail.com; Tel.: +886-958513101; Fax: +886-222454982

**Abstract:** Definition and imagination of an ideal city can be traced back to the origin of garden city in UK about 100 years ago. Since then, many different names and topics have been proposed and smart city is the one most recently proposed. Starting from 2000, more and more countries have developed various demonstration projects for the promotion of smart city in order to provide total solution for the promotion of sustainable development and social welfare. In fact, some of them have been successfully carried out. Some researchers in Taiwan argue that the current mechanism by which government subsidies are allocated for smart city demonstration projects warrants improvement. A comprehensive literature review determined that the development potential of smart cities should be prioritized in site selection for such demonstration projects. This study developed an evaluation framework on the basis of multi-criteria evaluation methods to enable the identification of suitable smart city demonstration sites. Evaluation criteria were first identified through the Delphi method. Next, the weights of each criterion were derived through the analytic hierarchy process. Furthermore, the capability of the proposed evaluation model was determined through simulation testing. Four demonstration sites are simulated, they are: Taipower Smart Community, Yinlin Technology University campus, Taichung Creative Cultural Park, and Asian New Bay Area in Kaohsiung, It is expected that the research findings in this thesis can be helpful to the future decision for the demo site selection of smart city.

**Keywords:** smart city; demonstration project; multi-criteria evaluation; Delphi method; analytic hierarchy process

## 1. Introduction

Definition and imagination of an ideal city can be traced back to the origin of garden city in UK about 100 years ago. Since then, many different names and topics have been proposed and smart city is the one most recently proposed. Starting from 2000, more and more countries have developed various demonstration projects for the promotion of smart city in order to provide total solution for the promotion of sustainable development and social welfare. In Taiwan, some researchers argue that the current mechanism by which government subsidies are allocated for smart city demonstration projects warrants improvement. A comprehensive literature review determined that the development potential of smart cities should be prioritized through the selection of sites for smart city demonstration projects. This study developed an evaluation framework on the basis of multi-criteria evaluation (MCE) methods to enable the identification of suitable demonstration sites. An evaluation framework was first developed through the literature review, after which evaluation criteria were identified through a review of smart city projects implemented in Europe. Moreover, the differences between the proposed evaluation framework and the current system were examined. The proposed evaluation framework was assessed through two steps. Evaluation criteria were first identified through the Delphi method. Next, the weights of each criterion were derived through the analytic hierarchy process. Furthermore,

the capability of the proposed evaluation model was determined through simulation testing. Simulations were performed for four demonstration sites Taipower Smart Community, National Yunlin University of Science and Technology, Taichung Creative Cultural Park, and the Asia New Bay Area in Kaohsiung (hereafter the Asia New Bay Area). The findings are expected to facilitate the selection of future smart city demonstration sites.

## 2. Theoretical Background

### 2.1. Smart City

Smart cities, the emergent future versions of a city, run in part on streams of data that continuously flow among physical objects, actors, and institutions that respectively define, inhabit, and govern cities [1–4].

The agenda of smart cities extends that of computationally networked urbanism, which has been in progress since the early 1970s. Smart cities have also been termed wired cities [5], cyber cities [6], digital cities [7], intelligent cities [8], and sentient cities [9], among others [10,11]. The concept of smart cities overlaps with other popular city framings (e.g., resilient cities, sustainable cities, safe cities, and ecocities). In contrast to earlier formulations of networked urbanism, smart cities—as a concept, ideal, and assemblage of products—rapidly gained traction in industry, government, and academia beginning in the late 2000s to become a global urban agenda [11,12].

The present international evaluation team is in collaboration with the Institute of Urban Mobility in Architecture and consists of three research units, one each from the Vienna University of Technology Regional Science Center, the Department of Geography of the University of Ljubljana, and the Delft University of Technology. The team believes that a smart city must be characterized by the balance of a smart economy, smart environment, smart people, smart governance, smart mobility, and smart living [13,14]. According to evaluation criteria presented by Cocchia (2014), community cohesion is a humanistic attribute of smart cities: "A Smart City is a city well performing built on the 'smart' combination of endowments and activities of self-decisive, independent and aware citizens." [15].

### 2.2. Smart City Projects

The concept of experimental cities, a living lab operates in a real-life context and adopts a user-centric approach in that users participate in the innovation and development of new business service models. In other words, users are present in every stage of product development and contribute to the innovation of a product or service by drawing from their own life or social experiences. Products or services developed in this manner satisfy the needs of the public and more closely align with its everyday realities. Studies have reported that user participation in the innovation development process of a product or service corresponds to a lower likelihood of market failure [16–18].

Smart city demonstration project sites are sites selected for experiments involving smart city models or smart service models. This can involve daily life services, infrastructure testing, and operating procedure simulations. The primary site selection criteria for such projects are the sustainable intelligentization needs associated with the sites and the site participants' willingness to cooperate with project implementation. Other selection criteria include basic infrastructure, as well as the sustainable smart service development of the sites and resources to be invested into the sites. By performing an integrated evaluation of the criteria, researchers can select the site that best match their objectives and needs [19–21].

For example, Genova is one of the more aged cities in Europe; citizens over 65 are the 27% of the inhabitants. It means a low awareness about the smart city idea and a low ICT education level. However, elder people are main stakeholders of smart city initiatives and services; for example, e-health systems, better public transport services, cheaper heating and cooling plants. Therefore, they should be educated and adequately informed and involved in the smart city projects and some not-for-profit members of GCSA are working just for this goal. Only with the higher active participation of all the citizens the smart city

could produce and deliver the higher public, economic and social value for all. Comparing the key players in Genoa with those in Amsterdam, both cities have a top–down process driven by a public body (i.e., the municipality). Smart city projects are complex, require careful planning and substantial funds. To obtain favorable results, the directions of smart cities' development must be defined. Amsterdam selected a hierarchical, closed governance model, whereas Genoa opted for a flat, open one. Genoa, which centers on smart initiatives involving infrastructure and less involved in digital initiatives, considers citizens and nonprofit organizations key players in the success of the project. Moreover, the democratic organization of GSCA contributes crucially to the development of the project and to the achievement of a higher consensus. Determining which solution is the optimal one is challenging; perhaps all solutions can be considered the optimal ones for the city for which they were designed. A comprehensive smart city project, which aims to transform the profile of a city, must be specific to that city and harmonized with its characteristics, whether cultural or otherwise [22].

Dameri (2014) used a formal organization/quadruple helix model to explore smart city governance in Genoa. Specifically, key actors and participation were adopted as variables to compare smart city governance in Genoa and Amsterdam. For Genoa, participation was set as open, whereas for Amsterdam, it was set as closed [22,23] (Table 1).

**Table 1.** Key actors in Genova and Amsterdam smart city.

| Demonstration Projects | Genova Smart City | Amsterdam Smart City |
|---|---|---|
| Starting process | Top-down | Top-down |
| Participation | Open | Closed |
| Structure | Flat | Hierarchical |
| First mover | Public body | Public body |
| Actors | Public, Private and Not-for-profit | Public–private partnership |
| Governance | Formal organization (Quadruple helix model) | Formal organization (Quadruple helix model) |

Resource from [22,23].

### 2.3. MCE

Multiple criteria decision-making (MCDM), which originated from Koopmans's concept of efficient vectors [24], has been used by various scholars and decision makers to address problems related to design, selection, and evaluation. Using multiple criteria as the basis of evaluation, decision makers express their preference structure. They then determine suitable solutions or the ordering of the alternative solutions in terms of their suitability. MCDM can help decision-makers analyze and rank a limited number of solutions according to their pros and cons, thereby determining an ideal solution [25,26]. Common MCDM methods include the simple multi-attribute ranking technique [27,28] and the analytic hierarchy process [29].

Among the numerous conflict management approaches available, MCDM is one of the most widely employed. Under MCDM, practical problems are often characterized by several incommensurable and competing (conflicting) criteria, and no solution satisfies all the criteria simultaneously. Thus, a compromise solution for such problems—that is, a feasible solution that closely approximates the ideal solution—must be determined to enable a final decision to be made. The MCDM procedure applied herein comprises the following steps: (a) Establishing system evaluation criteria relating system capabilities to goals; (b) developing (designing) alternative systems for attaining these goals (generating alternatives); (c) evaluating alternatives in terms of criteria (i.e., the values of criterion functions); (d) applying a normative multi-criteria analysis method (e.g., compromise ranking); (e) accepting one alternative as the optimal or preferred solution. In step, if the final solution is not accepted, new information must be collected, after which one proceeds to the next iteration of multi-criteria optimization.

Steps (a) and (e) are performed at the upper (decision) level, where decision-makers play a central role. The other steps are mostly mechanical tasks. Alternatives can be generated and their feasibility can be tested using mathematical models and physical models. This may entail experiments on the existing system or on other similar systems. Generating alternatives may be challenging and complex. No general procedure or model has been developed, and no mathematical procedure can replace the human creativity involved in generating and evaluating alternatives. In this process, constraints are seen as high-priority objectives and must be considered and satisfied. Assuming that each alternative is evaluated according to each criterion function, the compromise ranking method can be applied to determine a compromise solution, where a compromise refers to an agreement established through mutual concessions. The compromise solution maximizes utility for the majority of the group (in terms of the concordance measure $S$) and minimizes the regret of the opponent (in terms of the discordance measure $R$). VIKOR, the present compromise ranking method, is feasible to implement in MCDM. The VIKOR algorithm determines the weight stability intervals corresponding to the compromise solution, with the "input" weights determined by experts [30,31].

## 3. MCE Model Construction

### 3.1. Evaluation Criteria Framework

In the MCE of a smart city subsidization plan, the weights of various factors must be determined in advance. These factors include who (project implementation ability), what (project scope), where (smart community), and sustainability (financial self-reliance). According to international retrospective indicators and domestic predictive indicators, the MCE of smart city demonstration projects in this study involved five dimensions, namely environment, governance, mobility, people, and feedback, according to which evaluation indexes were configured (Table 2).

**Table 2.** MCE framework for evaluating smart city demonstration projects.

| Evaluation Dimension | A. Environment | | B. Governance | | C. Mobility | | D. People | | E. Feedback | |
|---|---|---|---|---|---|---|---|---|---|---|
| Index group | Indexes A-1 to A-3 | | Indexes B-1 and B-2 | | Indexes C-1 and C-2 | | Indexes D-1 and D-2 | | Indexes E-1 to E-4 | |
| | A-1 | 1–3 | B-1 | 1–2 | C-1 | 1–4 | D-1 | 1–3 | E-1 | 1–3 |
| Index label | A-2 | 1–3 | B-2 | 1–4 | C-2 | 1–2 | D-2 | 1–3 | E-2 | 1–4 |
| | A-3 | 1–3 | | | | | | | E-3 | 1–2 |

### 3.2. Key Factor Framework: Delphi Analysis of Expert Responses

To three cohorts of decision-makers: civil servants ($n = 8$), academics in relevant fields ($n = 5$), and industry professionals ($n = 9$), 22 copies of a questionnaire survey were administered. Thirty valid questionnaires were retrieved, corresponding to a response rate of 86%. The respondents in the second stage of the study worked in the following fields: urban planning, environmental engineering, civil engineering, and architectural engineering.

With reference to similar studies, this study used the average score and expert consensus ($G_i$) score as selection thresholds for the evaluation indexes. An evaluation index required an average score of 3.5 or above and a $G_i$ score of $\geq 3.0$ to be considered a key factor. After the survey responses were analyzed, the evaluation indexes that did not meet these requirements were removed from the key factor framework. Two rounds of surveys were conducted, and the Delphi method was used to analyze the responses and calculate the corresponding scores. If the verification ($Z_i$) score of a factor exceeded 0, it indicated expert consensus and the convergence of the factor scores [32]. Accordingly, the proposed MCE framework was established for site selection evaluation. The results from the Delphi analysis were subsequently applied to the weighting of the evaluation indexes (Table 3).

**Table 3.** Selection of evaluation indexes through the Delphi method.

| Evaluation Dimension | Evaluation Criterion | Evaluation Index | Relative Importance | | | Zi Verification Score | Gi Expert Consensus Score |
|---|---|---|---|---|---|---|---|
| | | | Maximum Score | Minimum Score | Average Score | | |
| Environment | Environmental site conditions | Site location | 5 | 3 | 4.2 | 0.81 | 4 |
| | | Site maintenance area | 5 | 1 | 3.4 | 1.38 | 3 |
| | | Level of environmental intelligentization | 5 | 2 | 3.9 | 1.06 | 3.5 |
| | Digitization | Digitization of environmental monitoring | 5 | 2 | 4 | 1.25 | 3.5 |
| | | Digitization of passive energy conservation | 5 | 1 | 4.2 | 0.68 | 3 |
| | | Improvement in quality of life through digitization | 5 | 2 | 4 | 1.06 | 3.5 |
| Governance | Improvability | Issues concerning sustainability improvement | 5 | 2 | 4.1 | 0.93 | 3.5 |
| | | Issues concerning intelligentization improvement | 5 | 2 | 3.9 | 1.06 | 3.5 |
| | Sustainability | Expenditure ratios of collaborative projects | 5 | 1 | 4.3 | 0.75 | 3 |
| | | Implementation of smart building projects | 5 | 2 | 3.8 | 1.25 | 3.5 |
| | | Implementation of basic smart infrastructure | 5 | 2 | 4 | 1 | 3.5 |
| | | Implementation of carbon reduction and energy conservation | 5 | 1 | 3.7 | 1.31 | 3 |
| Mobility | Informationization | Mobile phone penetration rate | 5 | 3 | 4.4 | 0.56 | 4 |
| | | Wi-Fi coverage | 5 | 3 | 4.6 | 0.38 | 4 |
| | | Mobile broadband (3G and 4G) usage penetration rate | 5 | 3 | 4.6 | 0.38 | 4 |
| | | Informationization performance | 5 | 3 | 3.9 | 1.13 | 4 |
| | Sustainable transportation | Public transportation usage rate | 5 | 1 | 4.3 | 0.69 | 3 |
| | | Nonmotorized vehicle usage rate | 5 | 2 | 4 | 1.06 | 3.5 |
| | | Renewable energy transportation usage rate | 5 | 3 | 3.8 | 1.06 | 4 |
| People | Humanistic qualities | Community cohesion | 5 | 3 | 4.5 | 0.5 | 4 |
| | | Community identity | 5 | 3 | 4.3 | 0.63 | 4 |
| | | Open-mindedness | 5 | 3 | 4.3 | 0.69 | 4 |
| | Lifelong learning | Digital management and training performance | 5 | 2 | 3.8 | 1.19 | 3.5 |
| | | Digital learning platform penetration rate | 5 | 3 | 3.9 | 1.13 | 4 |
| | | Employment growth rate attributable to digital management and training | 5 | 1 | 3.4 | 1.44 | 3 |
| Feedback | Maintenance and management systems | Environmental monitoring performance | 5 | 2 | 3.9 | 1.13 | 3.5 |
| | | Information technology application | 5 | 3 | 4.2 | 0.81 | 4 |
| | | Carbon reduction and energy conservation performance | 5 | 2 | 3.4 | 1.56 | 3.5 |
| | Potential contribution to urban development | Self-liquidating performance | 5 | 2 | 4.3 | 0.75 | 3.5 |
| | | Site proximity to city center | 5 | 2 | 4.1 | 0.88 | 3.5 |
| | | Cultural diversity | 5 | 2 | 3.7 | 1.31 | 3.5 |

According to the expert evaluation, the number of variables in the indexes was reduced from 12 to 10 and from 36 to 31 as follows. Pollution control and energy resources were merged into digitization. The expert consensus score of government financial benefits was deleted because it was lower than 3.5. Expenditure ratios of collaborative projects was moved to the second level of the sustainability index. The expert consensus score of the implementation rate of smart or green buildings was deleted because it was lower than 3.5. Site maintenance area and employment growth rate attributable to digital management and training were retained because their expert consensus scores both exceeded 3.6 in the first investigation.

### 3.3. Analysis of Factor Weights: Examination of Expert Responses through the AHP

Weights were assigned to the evaluation dimensions, evaluation criteria, and evaluation indexes to reflect their relative importance in the MCE framework of smart city demonstration projects. The 12 evaluation criteria were reduced to 10, and the original 36 evaluation indexes were reduced to 31. The evaluation criteria were paired with the evaluation dimensions and assigned weights. These weights were then referenced when the weights of the evaluation indexes were adjusted to optimize the MCDM model (Table 4).

**Table 4.** Analysis of factor weights for the MCE of smart city demonstration projects.

| Evaluation Dimension | Weight | Evaluation Criterion | Weight | Evaluation Index | Original Weight | Adjusted Weight | Ordering |
|---|---|---|---|---|---|---|---|
| A Environment | 0.186 | A1 Environmental site conditions | 0.116 | A1-1 Site location | 0.382 | 0.044 | 4 |
| | | | | A1-2 Site maintenance area | 0.251 | 0.029 | 19 |
| | | | | A1-3 Level of environmental intelligentization | 0.367 | 0.034 | 13 |
| | | A2 Digitization | 0.091 | A2-1 Digitization of environmental monitoring | 0.293 | 0.027 | 23 |
| | | | | A2-2 Digitization of passive energy conservation | 0.280 | 0.026 | 25 |
| | | | | A2-3 Improvement in quality of life through digitization | 0.427 | 0.039 | 8 |
| B Governance | 0.210 | B1 Improvability | 0.092 | B1-1 Issues concerning sustainability improvement | 0.489 | 0.045 | 3 |
| | | | | B1-2 Issues concerning intelligentization improvement | 0.511 | 0.047 | 1 |
| | | B2 Sustainability | 0.132 | B2-1 Expenditure ratios of collaborative projects | 0.236 | 0.032 | 16 |
| | | | | B2-2 Implementation of smart building projects | 0.224 | 0.030 | 18 |
| | | | | B2-3 Implementation of basic smart infrastructure | 0.321 | 0.043 | 5 |
| | | | | B2-4 Implementation of carbon reduction and energy conservation | 0.219 | 0.029 | 20 |
| C Mobility | 0.227 | C1 Informationization | 0.101 | C1-1 Mobile phone penetration rate | 0.312 | 0.032 | 15 |
| | | | | C1-2 Wi-Fi coverage | 0.277 | 0.028 | 21 |
| | | | | C1-3 Mobile broadband (3G and 4G) usage penetration rate | 0.229 | 0.023 | 27 |
| | | | | C1-4 Informationization performance | 0.182 | 0.019 | 29 |
| C Mobility | 0.227 | C2 Sustainable transportation | 0.069 | C2-1 Public transportation usage rate | 0.526 | 0.037 | 10 |
| | | | | C2-2 Nonmotorized vehicle usage rate | 0.264 | 0.018 | 30 |
| | | | | C2-3 Renewable energy transportation usage rate | 0.210 | 0.015 | 31 |

**Table 4.** *Cont.*

| Evaluation Dimension | Weight | Evaluation Criterion | Weight | Evaluation Index | Original Weight | Adjusted Weight | Ordering |
|---|---|---|---|---|---|---|---|
| D People | 0.210 | D1 Humanistic quality | 0.123 | D1-1 Community cohesion | 0.381 | 0.047 | 2 |
| | | | | D1-2 Community identity | 0.342 | 0.042 | 6 |
| | | | | D1-3 Open-mindedness | 0.277 | 0.034 | 12 |
| | | D2 Lifelong learning | 0.089 | D2-1 Digital management and training performance | 0.368 | 0.033 | 14 |
| | | | | D2-2 Digital learning platform penetration rate | 0.423 | 0.038 | 9 |
| | | | | D2-3 Employment growth rate attributable to digital management and training | 0.209 | 0.020 | 28 |
| E Feedback | 0.167 | E1 Maintenance and management systems | 0.117 | E1-1 Environmental monitoring performance | 0.226 | 0.027 | 24 |
| | | | | E1-2 Information technology application | 0.217 | 0.026 | 26 |
| | | | | E1-3 Carbon reduction and energy conservation performance | 0.259 | 0.031 | 17 |
| | | | | E1-4 Self-liquidating performance | 0.298 | 0.035 | 11 |
| | | E2 Potential contribution to urban development | 0.070 | E2-1 Site proximity to city center | 0.602 | 0.042 | 7 |
| | | | | E2-2 Cultural diversity | 0.398 | 0.028 | 22 |

## 4. Case Studies of Simulated MCE Applications

This study performed evaluation simulations at the following four sites: Taipower Smart Community, National Yunlin University of Science and Technology, Taichung Creative Cultural Park, and the Asia New Bay Area. The evaluation simulations were used to verify the feasibility of using the MCE model to assess smart city demonstration projects. Numerical data were also collected and used to evaluate site conditions, including with regard to resource availability. According to its performance, each site was reviewed, and recommendations were formulated to facilitate its development into a smart community. From the MCE simulation, an original performance score and an adjusted score, both between 0 and 1, were generated for each site on the basis of the weights assigned according to the evaluation criteria (Charts 1–4).

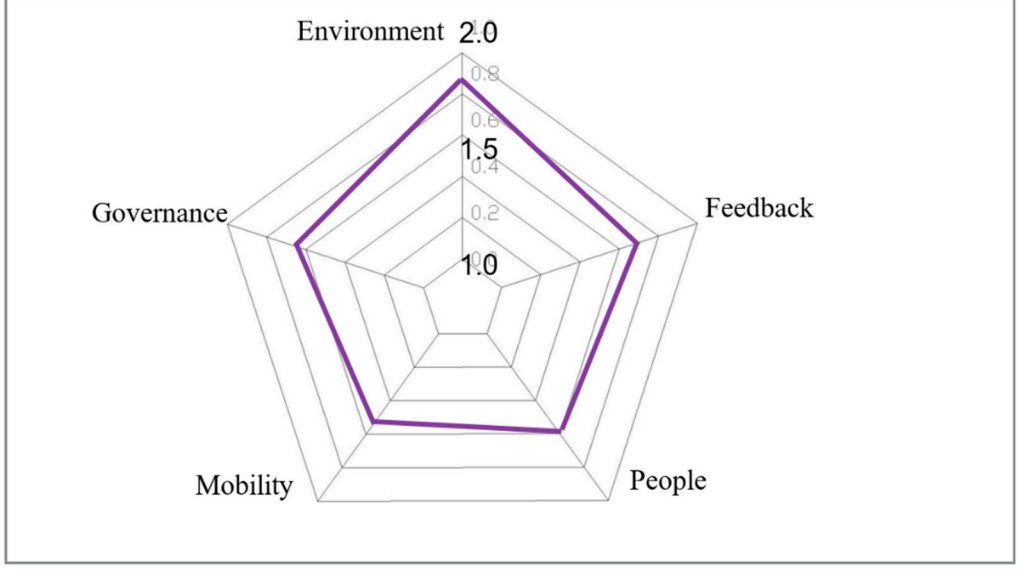

**Chart 1.** MCE results of the Taipower Smart Community site.

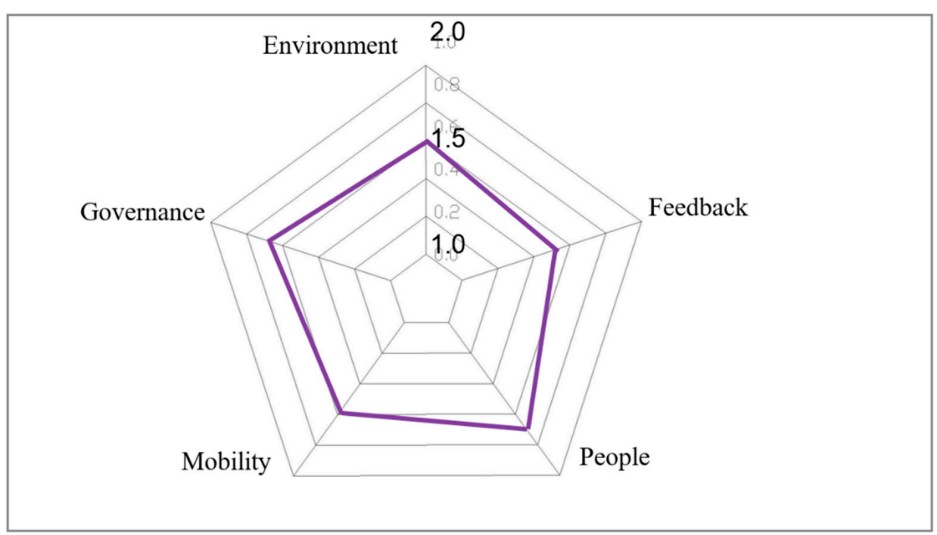

**Chart 2.** MCE results of the National Yunlin University of Science and Technology site.

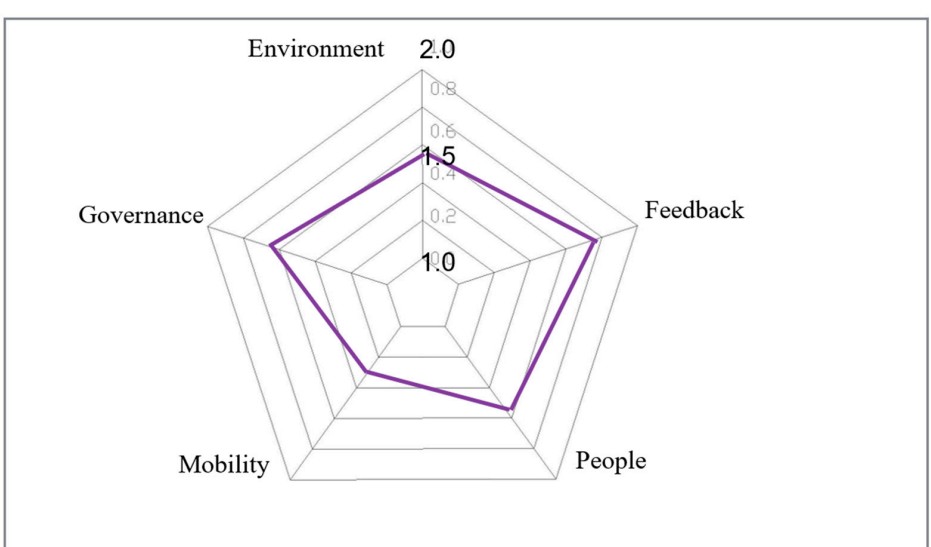

**Chart 3.** MCE results of the Taichung Creative Cultural Park site.

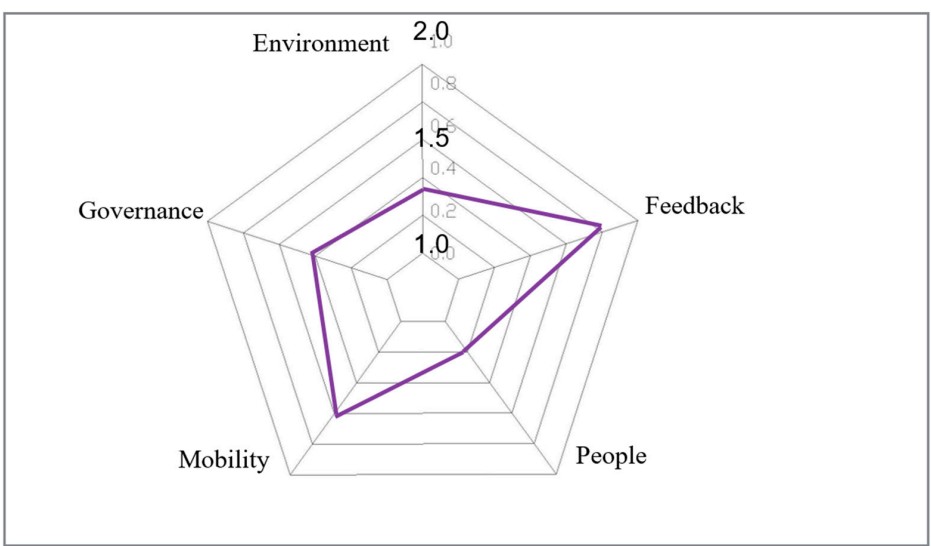

**Chart 4.** MCE results of the Asia New Bay Area site.

## 5. Conclusions

Some researchers in Taiwan assert that the current mechanism by which government subsidies are allocated for smart city demonstration projects must be improved. A comprehensive literature review revealed that the development potential of smart cities should be prioritized through the selection of sites for smart city demonstration projects. This study proposed an MCE model for smart city demonstration projects following a literature review and empirical analysis. The findings are expected to serve as a reference for future smart city–related projects. The key findings are summarized as follows:

### 5.1. Inclusion of Environment, Governance, Mobility, People, and Feedback in the MCE of Smart City Demonstration Projects

Environment: The environmental dimension contained two evaluation criteria, namely environmental site conditions and digitization. The evaluation indexes of the environmental site conditions were site location, site maintenance area, and level of environmental intelligentization. The evaluation indexes of digitization were digitization of environmental monitoring, digitization of passive energy conservation, and improvement in quality of through digitization.

Governance: The governance dimension contained two evaluation criteria, namely improvability and sustainability. The evaluation indexes of improvability were issues concerning sustainability improvement and issues concerning intelligentization improvement. The evaluation indexes of sustainability were the expenditure ratios of collaborative projects, implementation of smart building projects, implementation of basic smart infrastructure, and implementation of carbon reduction and energy conservation.

Mobility: The mobility dimension contained two evaluation criteria, namely informationization and sustainable transportation. The evaluation indexes of informationization were mobile phone penetration rate, Wi-Fi coverage, mobile broadband (3G and 4G) usage penetration rate, and informationization performance. The evaluation indexes of sustainable transportation were the public transportation usage rate, nonmotorized vehicle usage rate, and renewable energy transportation usage rate.

People: The people dimension contained two evaluation criteria, namely humanistic qualities and lifelong learning. The evaluation indexes of humanistic qualities were community cohesion, community identity, and open-mindedness. The evaluation indexes of lifelong learning were digital management and training performance, digital learning platform penetration rate, and employment growth rate attributable to digital management and training.

Feedback: The feedback dimension contained two evaluation criteria, namely maintenance and management systems and potential contribution to urban development. The evaluation indexes of maintenance and management systems were environmental monitoring performance, information technology application, carbon reduction and energy conservation performance, and self-liquidating performance. The evaluation indexes of potential contribution to urban development were site proximity to the city center and cultural diversity.

### 5.2. Importance of Evaluation Indexes for the MCE of Smart City Demonstration Projects

In descending order, the weights of the 10 evaluation criteria are as follows: sustainability, humanistic qualities, maintenance and management systems, site environmental conditions, informationization, improvability, digitization, lifelong learning, potential contribution to urban development, and sustainable transportation. Therefore, the expert panels concluded that the consideration of sustainability, humanistic qualities, and maintenance and management systems (under the governance, people, and feedback dimensions, respectively) should be prioritized in site selection. Sustainable transportation had a lower weight, indicating its lower importance. Given that sustainable transportation was under the mobility dimension, it suggested that mobility constituted a less essential consideration

in the early stages of site selection. This also demonstrated that high intelligentization was not a requirement for site selection.

In descending order, the weights of the 31 evaluation indexes are as follows: issues concerning intelligentization improvement, community cohesion, issues concerning sustainability improvement, site location, implementation of basic smart infrastructure, community identity, site proximity to city center, improvement in quality of life through digitization, digital learning platform penetration rate, public transportation usage rate, self-liquidating performance, open-mindedness, level of environmental intelligentization, digital management and training performance, mobile phone penetration rate, expenditure ratios of collaborative projects, carbon reduction and energy conservation performance, implementation of smart building projects, site maintenance area, implementation of carbon reduction and energy conservation, Wi-Fi coverage, cultural diversity, digitization of environmental monitoring, environmental monitoring performance, digitization of passive energy conservation, information technology application, mobile broadband (3G and 4G) usage penetration rate, employment growth rate attributable to digital management and training, informationization performance, nonmotorized vehicle usage rate, and renewable energy transportation usage rate. The higher weights of intelligentization improvement, community cohesion, issues concerning sustainability improvement, site location, implementation of basic smart infrastructure, and community identity indicated that the expert panels considered them more essential. These evaluation indexes corresponded to the following evaluation criteria: improvability, humanistic qualities, site environmental conditions, and sustainability. Less essential indexes corresponded to the evaluation criteria of informationization and sustainable transportation, both of which were under the mobility dimension. This once more indicated that the experts considered mobility less essential in the early stages of site selection, and that high intelligentization was not a requirement for site selection.

The analysis of survey responses revealed that the three expert panels concurred that high intelligentization was not an essential requirement for site selection. This consensus accords with the present premise that the development potential of smart cities should be prioritized in site selection.

### 5.3. Development of an Objective, Quantifiable Evaluation Model Using MCE Simulations

In descending order, the intelligentization performance of the simulation sites the first year of evaluation is as follows: Taipower Smart Community, National Yunlin University of Science and Technology, Taichung Creative Cultural Park, and the Asia New Bay Area. Their development potential is in reverse order; the Asia New Bay Area had the highest potential, followed by Taichung Creative Cultural Park, National Yunlin University of Science and Technology, and Taipower Smart Community. Taipower Smart Community, the Asia New Bay Area, Taichung Creative Cultural Park, and National Yunlin University of Science and Technology is the descending order of the projected benefits of the sites, as indicated in an analysis of their establishment durations and government funding ratios.

According to the simulation results, high intelligentization corresponded to the highest potential for becoming a smart community demonstration site (e.g., Taipower Smart Community). However, if such a community, such as National Yunlin University of Science and Technology, lacks long-term planning and effective management, it will not be a suitable demonstration site. If they lack basic smart infrastructure and have high goals for self-liquidation, and assuming rapid growth in intelligentization within the next 2 years and a high government funding ratio, sites—in this case, the Asia New Bay Area and Taichung Creative Cultural Park—can be projected to have the highest potential to become smart communities.

The results serve as a reference for relevant units in the determination of subsidy allocation for smart community development in Taiwan. An MCE must comprehensively assess various factors to inform funding agencies of the expected benefits of the financial support they provide, on the basis of which they can adjust or reallocate the funding for the

following fiscal year. Such an evaluation mechanism forms a virtuous cycle that encourages the participation of certain stakeholders (i.e., private sector operators). The findings are expected to facilitate the selection of future smart community demonstration sites.

**Author Contributions:** Conceptualization, M.-S.S. and S.-G.S.; methodology, Y.-H.P. and M.-S.S.; software, M.-S.S.; validation, M.-S.S., S.-G.S. and Y.-H.P.; formal analysis, M.-S.S.; investigation, M.-S.S.; resources, M.-S.S.; data curation, M.-S.S.; writing—original draft preparation, M.-S.S.; writing—review and editing, Y.-H.P. and M.-S.S.; visualization, M.-S.S.; supervision, M.-S.S.; project administration, M.-S.S. All authors have read and agreed to the published version of the manuscript.

**Funding:** This research received no external funding.

**Institutional Review Board Statement:** Not applicable.

**Informed Consent Statement:** Not applicable.

**Data Availability Statement:** Data openly available in a public repository. The data that support the findings of this study are openly available in Preprints [10.20944/preprints202112.0351.v1].

**Acknowledgments:** We like to thank Shih, S.-G., Perng Y.-H. of NTUST and Architecture and Building Research Institute, Ministry of the Interior. Other, John C.-Y.L. in NTU for providing data and for valuable discussions on earlier versions of this paper.

**Conflicts of Interest:** The authors declare no conflict of interest.

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
