# Peer review of "Multi-Criteria Evaluation of Site Selection for Smart Community Demonstration Projects"

_smartcities, doi:10.3390/smartcities5010002_

Round 1
Reviewer 1 Report
While the idea of the research itself is novel and provides valuable insight to its domain, there are several issues that should be improved,
*The rationality behind the choice of the specific parameters chosen within each component is unclear, for example, in the governance component, is improvability and sustainability components of governance, what is the reference to that and why were these two specific parameters chosen?
*The same goes for the rest of the five dimensions, there are many more parameters to choose from and that may enrich the idea behind the model, especially in both the environmental component and the community participation and perception one (what is the reference for the use of the term "humanistic qualities"), community cohesion does not confer to humanistic attributes)
*Lines 74-81 are not clear, the comparison between genoa and Amesterdam, what influenced the open and closed classification and where does this fit within the research?
*the reasons for reducing the variables from 12 to 10 and from 36 to 31 should be stated, to give depth to the study.
*The conclusion part introduces the results and discussion, although it should only state the kep concluding remarks of the research, kindly rewrite the results and discussion and add the key takebacks and conclusions to the conclusion section.
*the tables at the end should be embedded within the structure of the paper and discussed further.
Overall, the fieldwork itself needs to be better emphasized and explained, the research needs elaboration and a consistent results and discussion section that explains the use and phases of the model.
Author Response
Response to reviewer 1
(please read green color words in the main text)
While the idea of the research itself is novel and provides valuable insight to its domain, there are several issues that should be improved,
Point 1*The rationality behind the choice of the specific parameters chosen within each component is unclear, for example, in the governance component, is improvability and sustainability components of governance, what is the reference to that and why were these two specific parameters chosen?
Responde 1: Thanks for your kindest suggestion.I have expained the reason of improvability and sustainability components of governance in the governance component in the ch3.1.
Point 2*The same goes for the rest of the five dimensions, there are many more parameters to choose from and that may enrich the idea behind the model, especially in both the environmental component and the community participation and perception one (what is the reference for the use of the term "humanistic qualities"), community cohesion does not confer to humanistic attributes)
Responde 2: Thanks for your kindest suggestion. I have added background information in the bottom of ch2.1.
Point 3*Lines 74-81 are not clear, the comparison between genoa and Amesterdam, what influenced the open and closed classification and where does this fit within the research?
Responde 3: Thanks for your kindest suggestion. I have added background information of the comparison between Genoa and Amesterdam in the bottom of ch2.2.
Point 4*the reasons for reducing the variables from 12 to 10 and from 36 to 31 should be stated, to give depth to the study.
Responde 4: Thanks for your kindest suggestion. I have added the process for reducing the variables from 12 to 10 and from 36 to 31 in the ch3.2.
Point 5*The conclusion part introduces the results and discussion, although it should only state the kep concluding remarks of the research, kindly rewrite the results and discussion and add the key takebacks and conclusions to the conclusion section.
Responde 5:Thanks for your kindest suggestion. I have rewritten a consistent results in the top of ch5. and more discussion section in the bottom of the ch5.
Point 6*the tables at the end should be embedded within the structure of the paper and discussed further.
Responde 6:Thanks for your kindest suggestion.the tables at the end have be embedded within the structure of the paper and discussed further.
Overall, the fieldwork itself needs to be better emphasized and explained, the research needs elaboration and a consistent results and discussion section that explains the use and phases of the model.
Thanks for the precious suggestions from the reviewer again.
Submission Date
26 November 2021
Date of this review
06 Dec 2021 14:01:30

Reviewer 2 Report
1.The paper describes an evaluation framework based on multi-criteria evaluation methods. The framework is targeted to the analysis of suitable smart community demonstration sites.
2.Evaluation criteria are identified through the Delphi method. The analytic hierarchy process is used as the basic multicriteria approach.
3.The capability of the proposed evaluation approach was determined through simulation testing.
4.The examination of the evaluation process is based on four alternatives: (i) Taipower Smart Community, (ii) National Yunlin University of Science and Technology, (iii) Taichung Creative Cultural Park, and (iv) the Asia New Bay Area.
5.The reference 21 contains the type mistake (“T.L. Saaty” is correct).
6.Generally, the paper can be accepted after minor revision. It is reasonable to look for the publications of Prof. Gwo-Hshiung Tzeng (many materials on multicriteria applications including applications in civil engineering and urban studies).
Author Response
Response to reviewer 2
(please read blue color words in the main text)
Point1.The paper describes an evaluation framework based on multi-criteria evaluation methods. The framework is targeted to the analysis of suitable smart community demonstration sites.
Responde 1: Thanks for your kindest recognition for the value of the study.
Point 2.Evaluation criteria are identified through the Delphi method. The analytic hierarchy process is used as the basic multicriteria approach.
Responde 2: Thanks for your kindest recognition for the value of the study.
Point 3.The capability of the proposed evaluation approach was determined through simulation testing.
Responde 3: Thanks for your kindest recognition for the value of the study.
Point 4.The examination of the evaluation process is based on four alternatives: (i) Taipower Smart Community, (ii) National Yunlin University of Science and Technology, (iii) Taichung Creative Cultural Park, and (iv) the Asia New Bay Area.
Responde 4: Thanks for your kindest recognition for the value of the study.
Point 5.The reference 21 contains the type mistake (“T.L. Saaty” is correct).
Responde 5: Thanks for your kindest mark “T.L. Saaty” is correct. I have corrected the reference 26(now 26, before 21).
Point 6.Generally, the paper can be accepted after minor revision. It is reasonable to look for the publications of Prof. Gwo-Hshiung Tzeng (many materials on multicriteria applications including applications in civil engineering and urban studies).
Responde 6: Thanks for your kindest offering of the publications of Prof. Gwo-Hshiung Tzeng. the auther gained a lot. I have added background information in the ch2.3.
Submission Date
26 November 2021
Date of this review
29 Nov 2021 20:21:46

Round 2
Reviewer 1 Report
Dear Authors,
Thank you for taking my comments into consideration, I am somewhat satisfied with the changes as well as the explanations given, however, please take the time to check the added paragraphs for language and spelling mistakes; (there are several instances of "have be", that should be "have been")
This manuscript is a resubmission of an earlier submission. The following is a list of the peer review reports and author responses from that submission.